From attributes to communities: a novel approach in social network generation

Uludağlı Muhtar Çağkan cagkan.uludagli@ieu.edu.tr m.c.uludagli@gmail.com
Oğuz Kaya
Department of Computer Engineering, İzmir University of Economics , İzmir , Turkey
Galán José Manuel
Electronic publication date: 2024 Nov 22
Publication date: 2024
Volume: 10
Electronic Location ID: e2483
Received 2024 Jul 23; Accepted 2024 Oct 16
Copyright: ©2024 Uludağlı and Oğuz
Copyright year: 2024
Copyright holder: Uludağlı and Oğuz
License: This is an open access article distributed under the terms of the Creative Commons Attribution License, which permits unrestricted use, distribution, reproduction and adaptation in any medium and for any purpose provided that it is properly attributed. For attribution, the original author(s), title, publication source (PeerJ Computer Science) and either DOI or URL of the article must be cited.
License URL: https://creativecommons.org/licenses/by/4.0/

Keywords: Graph generation, Node attributes, Social networks, Community

Funding: The authors received no funding for this work.

==============================
Generating networks with attributes would be useful in computer game development by enabling dynamic social interactions, adaptive storylines, realistic economic systems, ecosystem modelling, urban development, strategic planning, and adaptive learning systems. To this end, we propose the Attribute-based Realistic Community and Associate NEtwork (ARCANE) algorithm to generate node-attributed networks with functional communities. We have designed a numerical node attribute-edge relationship computation system to handle the edge generation phase of our network generator, which is a different method from our predecessors. We combine this system with the proximity between nodes to create more life-like communities. Our method is compared against other node-attributed social network generators in the area with using both different evaluation metrics and a real-world dataset. The model properties evaluation identified ARCANE as the leading generator, with another generator ranking in a tie for first place. As a more favorable outcome for our approach, the community detection evaluation indicated that ARCANE exhibited superior performance compared to other competing generators within this domain. This thorough evaluation of the resulting graphs show that the proposed method can be an alternate approach to social network generators with node attributes and communities.

Introduction

Artificial intelligence has gained incredible momentum in recent years, and in the context of computer games, one of its purposes is to create the illusion of interaction with real entities. The more the players immerse themselves into a computer-generated world, the richer their experience becomes. The immediate recollection for such game entities is the non-player characters (NPCs), the autonomous agents which are not controlled by the player but the computer (Uludağlı  & Oğuz, 2023). NPCs in proximity are usually related or have some kind of interaction that may involve scripted quests. These interactions lead to networks of NPCs, which can be traditionally represented by graphs (Newman, 2018).

While the graph is a fundamental data type in computer science, network science is a highly active interdisciplinary field and has seen heightened interest with the emergence of online social networks (Watts, 2004; Tabassum et al., 2018). The study of social networks becomes interesting when these large graphs are analyzed for communities, which is a subgraph where nodes have edges because they have common properties. For example, a subgraph in a social network might represent people with an interest in science fiction movies (Olson & Neal, 2015). Formally, we define community as a group of nodes that have higher probability of connection within that group than without (Fortunato & Hric, 2016). When this definition is based on the connectivity patterns of the graph, it is referred as a structural community, and if it is defined according to a common function or a role, then it is referred as a functional or meta community (Yang & Leskovec, 2012; Bonifati et al., 2020).

The edges between nodes can be formed for several reasons, such as proximity, a shared property, or a physical connection. They may also get connected because of their internal properties. It is possible to add these properties to the definition of nodes. These properties are called attributes, and such networks are called node-attributed networks (Chunaev, 2020). We can use these attributes to help us understand why two nodes have a connection. It is possible that two nodes are connected because they are similar with respect to their conceptual distance aspect.

We can also use attributes to procedurally generate a network where the connections between nodes rely on the agreement of similar attribute values. We can think of this as another case of procedural content generation for computer games and simulations (Shaker, Togelius & Nelson, 2016).

In this study, we propose an attribute-based realistic community and associate network algorithm, shortly named ARCANE, which generates a node-attributed network with functional communities. These communities are defined by the node attributes, and the edges between the nodes are created based on a combination of node attributes and their conceptual node positions. ARCANE rests on the idea that attributes and conceptual distance between the attributes are a good indication of connection in real world networks (Newman, 2018). The algorithm is novel in its definition of generalized attributes and the generalized functions to create connections between nodes. Many parameters of the graph generation process can be tweaked and adjusted to meet the requirements of the users.

Generating networks with attributes would be useful to study the common properties of social networks, but also in some aspects of computer game development such as creating more realistic and immersive content for computer games by enabling dynamic social interactions, adaptive storylines, realistic economic systems, ecosystem modeling, urban development, strategic planning, and adaptive learning systems. This study opens new frontiers in procedural content generation for computer games by creating these life-like communities.

The following section provides an overview of existing studies in the literature. Method section presents our assumptions and explains our method. Results and Discussion section gives our results with respect to the comparisons with similar or alternate methods and a real dataset. The final section concludes our article with our final thoughts and the future work that can be done to improve our method.

Related Work

Graph generation is a large field and there are comprehensive surveys that cover it (Chakrabarti & Faloutsos, 2006; Bonifati et al., 2020; Xiang et al., 2021). There are also different models for network generation, such as random graphs, power-law degree distributed random graphs, small-world model or exponential random graphs. These models are analyzed in the works of Aiello, Chung & Lu (2000), Aiello, Chung & Lu (2001) and Nobari et al. (2011).

It is also important to consider what a community is or how it is defined in different contexts. The following studies are comprehensive surveys that define the basic concepts, such the definitions of communities, community structure in real networks and overlapping communities (Fortunato, 2010; Fortunato & Hric, 2016). Some of the community generation methods are developed to assess the performance of community detection method in networks (Lancichinetti & Fortunato, 2009b; Orman & Labatut, 2009; Javed et al., 2018).

A notable network generator used for community detection in networks is the Lanchichinetti-Fortunato-Radicchi (LFR) benchmark, which is named after its authors (Lancichinetti, Fortunato & Radicchi, 2008). They wanted to improve the limitations of the earlier network generator by Girvan & Newman (2002), and incorporated the power-law for the degree distributions of the nodes to generate their network. This property can be found in real-world networks which can be named as scale-free networks. Being a scale-free network means that the few nodes have many edges and many nodes have few edges in the network (De Solla Price, 1965; Barabási & Albert, 1999). The power-law distribution is used for both the node degrees and the community creation to simulate the real-world network characteristics better.

These studies do not employ node attributes for generating the networks. There are studies that use node attributes but do not take communities into account. For example, DataSynth framework generate property graphs by using their novel graph partitioning algorithm (Prat-Pérez et al., 2017) and Attributed Graph Model (AGM) utilizes generative graph model (Leskovec et al., 2010) to create node attributes from previously observed networks (Pfeiffer et al., 2014).

One of the first studies that generates a network with both community structure and node attributes is called LFR-EA (Elhadi & Agam, 2013) which extends the LFR benchmark. The attribute generation over LFR is controlled by the total attribute count, domain values count for each attribute, and an assignment influence parameter.

Largeron et al. (2015) introduce the Attributed Graph Generator with Community Structure (ANC) algorithm, and it makes use of the concept of homophily which directs the nodes to form connections with the nodes that are more similar with respect to their attributes (McPherson, Smith-Lovin & Cook, 2001). They designed a network generation model featuring local preferential attachment, small-world properties, community structure, community homogeneity, and homophily. The node attributes are generated with normal distribution according to the total number attributes and a set of attribute descriptors with a standard deviation. Benyahia et al. (2016) developed the DANCer framework which extends the ANC algorithm for dynamic networks to incorporate the constant change in social networks.

Nettleton (2016) created a generator for online social networks. Although they did not generate the actual network, they generated suitable datasets for predetermined graph topologies with using different node attribute values, community structures, and data distributions.

Maekawa et al. (2019) presented acMark, a network generator with cluster labels and node attributes that employs a Bayesian approach. They state that this method has benefits such as flexible control on cluster separability, various distribution assignments for attribute values, node degrees and cluster sizes, and linear time complexity proportional to the generated edges. Maekawa et al. (2023) introduced GenCAT as an extension to acMark to overcome the problem of unrealistic simulation of the relations between the labels, node attributes, and graph topology, that can occur in previous generators.

Wang, Wang & Feng (2021) asserted that their FastSNG method is the fastest social network generator with attributes and communities. They did not give detailed examples about attribute generation process other than stating that it adopts the property graph model, and it has both categorical values and numerical values for the attributes.

Citraro & Rossetti (2021) presented X-Mark, one of the recent network generators that performs generation operation with using a combination of the structural topology and the node attributes. Their method used both categorical and continuous attribute value types, and it provided both the community homogeneity and the homophily principle in the resulting network.

Table 1 lists all the generators and their properties that have been reviewed in this section. In contrast to all the generators that have been discussed so far, we propose a novel method that forms the graph edges with the help of node attributes and conceptual proximity concepts at the same time.

Table 1 Existing graph generators. ✓ indicates that the given generator includes that property, while × indicates that it is not.

Comm. stands for “Communities”, Attr. stands for “Node attributes”, and Param. stands for “Adjustable method parameters/properties”.

The generator	Comm.	Attr.	Param.	Degree distribution	Properties	
G.&N. (Girvan & Newman, 2002)	✓	×	×	constant	one of the first generators	
LFR (Lancichinetti, Fortunato & Radicchi, 2008)	✓	×	×	power-law	power-law degree dist.	
Overlapp. LFR (Lancichinetti & Fortunato, 2009a)	✓	×	×	power-law	overlapping communities	
DataSynth (Prat-Pérez et al., 2017)	×	✓	✓	schema-driven	topology created with synthetic data	
AGM (Pfeiffer et al., 2014)	×	✓	✓	attribute-correlated	property-to-node matching algorithm	
LFR-EA (Elhadi & Agam, 2013)	✓	✓	✓	power-law	LFR with attributes	
ANC (Largeron et al., 2015)	✓	✓	✓	clustering method-driven	categorical attributes	
DANCer (Benyahia et al., 2016)	✓	✓	✓	clustering method-driven	dynamic graph & communities	
acMark (Maekawa et al., 2019)	✓	✓	✓	uniform, normal, power-law	class-preference probability	
FastSNG (Wang, Wang & Feng, 2021)	✓	✓	✓	power-law	claimed to be the fastest	
X-Mark (Citraro & Rossetti, 2021)	✓	✓	✓	power-law	one of the first attribute-related edge generation	
GenCAT (Maekawa et al., 2023)	✓	✓	✓	power-law, normal, input list	class-preference probability	
ARCANE	✓	✓	✓	power-law (adjustable)	modifiable distributions, object-oriented, attribute & proximity-related edge generation	

Method

The novelty of ARCANE lies in its ability to mimic the real-life formation of social networks. Generally, people meet because they are in proximity and maintain their connection if they have similar interests (Moustakas, 2023). While internet has provided another way to access people without being physically close, this is still how our network of relations are formed; people we know and interact are set by our physical distribution in the cities we live (Lesser, Fontaine & Slusher, 2000). The cities are formed on physical terrains which are formed by geological processes. One of the most used algorithms to generate terrains is the fractal called diamond-square algorithm (Fournier, Fussell & Carpenter, 1982). ARCANE initializes the distribution of nodes using the values in the heightmap generated by this fractal.

In addition to physical proximity, ARCANE makes use of node attributes to create edges between nodes. Each node is assigned the same number of attributes where the value of each attribute is randomly decided using a normal distribution. Using both the physical proximity and the agreement between node attributes, the algorithm decides whether there will be a connection between two nodes. This broad view of the algorithm can also be seen in Fig. 1. This section provides both details and rationale for each of these steps.

Figure 1 The overview of the method.

The algorithm starts by (A) generating a heightmap using the diamond-square algorithm. It then uses the values in the heightmap to (B) generate candidate positions. The next step is (C) to generate nodes using the candidate positions and the attributes. Once the nodes have their positions and their attributes, the algorithm (D) connects nodes using their conceptual and attribute similarity values.

Required graph properties

The generated graph will represent a social network; therefore, it should have the following set of properties.

Property 1: Proximity for forming relations. Real communities are created in real places, whether it is an offline or an online location (Lesser, Fontaine & Slusher, 2000). We can refer these locations as conceptual positions, since these locations can be determined according to being in a workplace, in a professional seminar, or in a friendly gathering.

Property 2: Randomness of node placements. To provide the locality of the nodes, our method employs the diamond-square algorithm to distribute the nodes pseudo-randomly into the graph. The diamond-square algorithm is a method mostly used for generating heightmaps for three-dimensional terrains (Fournier, Fussell & Carpenter, 1982). The algorithm is a commonly used tool in computer graphics to create realistic landscapes.

Most of the graph generators we have surveyed in our study do not consider creating the topology of the generated graph by a known method. They use the pre-developed graph topologies to draw their graphs without considering the effects of the node placements. However, in a real network, the positions of the nodes can be important for various aspects such as connectivity and reachability, community detection, network efficiency, and geographical constraints. Therefore, we assume that using a proven algorithm for a similar purpose can be beneficial for placing the nodes in our graph.

Property 3: Homophily principle. Homophily is best described as creating more edges between similar nodes, by assuming that the similar nodes have more common features for being connected (McPherson, Smith-Lovin & Cook, 2001). This principle is vital for generating synthetic networks that resemble real network properties. We ensure that our generated networks use the homophily principle by forming the edges between the nodes with the help of the similarity of node attributes.

Property 4: Node attributes with values for attraction. There are existing studies that generate networks with node attributes as have been previously covered in Related Work section. Although these examples contain node attributes, most of them did not give importance to the idea of the negative and positive attributes, and the effect of these attribute alignments on the edge formation process.

Property 5: Edge generation with power law degree distribution. Most real-world networks share common properties. One of these properties is power law degree distribution (Newman, 2018). Some existing network generators, specifically for social networks, use it for both for node and edge degrees, while some of the examples use it on either one of them. Since we use a different approach on node generation phase, we decided to use power law degree distribution only on our edge generation phase. In our method, the nodes that are the most similar and that have the least conceptual distance to other nodes create more edges, and the nodes that are less similar and that have more distance to other nodes create fewer edges.

Parameters

Table 2 lists the five direct parameters that can be used to generate a network using ARCANE. The number of nodes, N, and the number of attributes for each node, A, are used to create nodes for the graph G that will be created when the algorithm is complete. The parameter k determines the maximum number of nodes a cell can hold before it can be determined how far the nodes can reach to create an edge and if their conceptual distance is within a threshold before an edge is created using the maximum distance, L, and threshold for proximity, t.

Table 2 Description of generation parameters.

Parameter	Description	
N	Number of nodes	
A	Number of general attributes	
k	Maximum number of nodes in a cell	
L	Maximum distance	
t	Threshold for proximity	

While these parameters can be used to adjust for a specific context, ARCANE can be customized even further. The diamond-square algorithm that is used to create a heightmap has a roughness parameter (ρ) to determine how steep the change between neighbor positions will be. Setting ρ to values closer to 1 creates larger changes between neighbor nodes. The algorithm also allows the function to be used for the generation of attribute values. While a normal distribution is used by default, it is possible to change the probability distribution to fit the requirements of a specific context.

Algorithm

The initial step in the method, as shown in Fig. 1, is to use the diamond-square algorithm to create a heightmap that will be used to distribute the initial candidate positions for the nodes in the graph. This algorithm requires a grid of size (2n + 1)2 because it updates the midpoints as it operates on smaller sizes of grids. Therefore, the size of the grid is set to m × m where m is bound to the number of nodes N as given in Eq. (1). (1) m=2log2N−1+1.

This approach creates cells greater than or equal to the number of nodes. Each cell will have some number of candidate positions with respect to the value that has been generated by the diamond-square algorithm for that cell.

The diamond-square algorithm that operates on a grid of size m × m briefly runs as follows. Initially, the four cells at the corners are set to random values. The algorithm then alternates between the diamond and square steps. In the diamond step, the grid is treated as a square, and the midpoint that lies on the diagonals of the four corners of the square is set to the sum of the average value of the corners and a random value. In the diamond step, the nodes that form a diamond is considered and the midpoint that lies in the vertical and horizontal axes of this diamond is set to the sum of the average corners of the diamond and a random value. These steps are performed sequentially in smaller squares and diamonds until all cell values have been set.

During realization of the diamond-square algorithm, the values are normalized to integers between 1 and k so that the next step can place a list of candidate positions to each cell. The variable k can be set to any integer value where k > 1 to parameterize the generation of the network.

In computer graphics, a heightmap represents a terrain where higher cell values represent higher elevations. ARCANE treats higher values in the cells as more crowded and places candidate positions around the center of each cell randomly using a uniform distribution. These positions are kept in a list P and since k ≥ 1, the number of candidate positions in P are greater than the number of nodes.

In the networks generated by ARCANE, the nodes can have A number of attributes where each attribute has an affinity level. The algorithm allows the values for these affinity levels to be set using a custom function so that it is possible to use any required probability distribution. The default function for the affinity levels employs a normal distribution with μ = 0 and σ2 = 1. These attributes are placed into a list α of size A where each attribute is denoted with an integer subscript of α. Each node is assigned such a list to decide the affinity levels of their attributes.

A graph is commonly denoted as G = (V, E) where V is the set of nodes and E is the set of edges. For each node that will be in a generated graph G, a position is used sequentially from P, the list of candidate positions, and a list of attributes is generated. The edges between two nodes v1 ∈ V and v2 ∈ V is determined by using a combination of their attribute similarity, s(v1, v2), and conceptual proximity, d(v1, v2).

The attribute similarity s(v1, v2) is calculated by considering all the attribute affinity values in each node. As previously mentioned, the integer subscript i in αi denotes an attribute in list α. This attribute exists in both v1 and v2 since each node has the same number of attributes. The similarity is then (2) sv1,v2= ∑iAv1αi×v2αi

where v1(αi) denotes the affinity level value of the attribute αi for node v1, and v2(αi) does the same for node v2. If the result is greater than zero, then it can be said that the nodes are similar. Otherwise, the nodes are said to be dissimilar. It is apparent that when two affinity level values have the same sign, their product will be positive. This helps the algorithm to incorporate likes and dislikes during edge generation.

The conceptual proximity is a measure whether the nodes in question are close enough to form an edge. Since a heightmap is used to distribute node positions, Euclidean distance is employed to find the distance between them. To make it work, the algorithm requires two values; L, the maximum distance between two nodes to consider an edge, and t, the threshold value to determine if an edge will be formed. Using these values, the conceptual proximity is (3) dv1,v2=|pv1−pv2|L

where p(v) denotes the position of a node v. If d(v1, v2) ≤ t, then it can be said that the nodes are conceptually close. The threshold value can be determined with respect to the maximum distance L. In existing literature, there is no consensus on how small the threshold value should be. In our examples, we have set the threshold value to be t = 0.1⋅L, however, in different contexts the threshold value can be adjusted accordingly.

The algorithm requires s(v1, v2) > 0 and d(v1, v2) > t to form an edge between nodes v1 and v2. However, it must conform to the power law degree distribution when distributing these edges. The procedure is applied by creating a list of numbers that starts with the number of nodes, and values obtained by dividing it by two until the last value is one.

The power-law degree distribution in edge generation is ensured as follows. If there are 16 generated nodes, then the divided numbers in order are 8, 4, 2 and 1. The sum of these values is 15. The first number, 8, is increased by one to 9, so that their sum equals to the number of nodes. The list then includes {9, 4, 2, 1}. This means that 1 node should have 9 edges, 2 nodes should have 4 edges, 4 nodes should have 2 edges, and 9 nodes should have a single edge. This procedure is used in the same manner for different number of nodes.

The pseudocode for the grid initialization phase of our algorithm is given in Algorithm 1 , the one for the candidate position generation phase is given in Algorithm 2 , and the one for the overall graph generation phase is given in Algorithm 3 .

___________________________________________________________ Algorithm 1 Grid Initialization using the diamond-square algorithm. N controls the size of the grid, and k is used to scale grid values into the range [1,k]___________________________________________________________________   1:  Input: N, k  2:  Output: A matrix of size m × m  3:  m ← (2N) + 1   4:  grid ← 2D list of size m × m initialized to zero   5:  max ← m − 1   6:  grid ← diamond_square(grid, max, ρ)                                                                                                                                                                     ⊳ ρ can be used to adjust the roughness   7:  grid ← normalize(grid, k)   8:  return grid___________________________________________________________________________________________________________________________________________________________________________________________________________________________________________________________

________________________________________________________________________________________________________________________________________________________________________________________________________________________________________________________________________________ Algorithm 2 Generate candidate positions______________________________________________________________________________________________________________________________________________________________________________________________________________________   1:  Input: grid   2:  Output: List P of candidate positions   3:  P ← empty list   4:  for each cell in grid do  5:       (x,y) ← coordinates of cell   6:       cellValue ← grid[x][y]   7:       for i ← 1 to cellValue do  8:            randomX ← x + uniform(−0.5,0.5)   9:            randomY ← y + uniform(−0.5,0.5) 10:            add (randomX,randomY) to P 11:       end for 12:  end for 13:  return P________________________________________________________________________________________________________________________________________________________________________________________________________________________

______________________________________________________________________________________________________________________________________________________________________________________________________________________________________ Algorithm 3 Create the graph by employing the power law edge distribution. P is the list of candidate positions._________________________________________________________________________________________________________________________  1:  Input: N, A, L, t, P  2:  Output: Graph G  3:  V ← empty set                                                                                                                                                                                                                                                          ⊳ Set of nodes  4:  E ← empty set                                                                                                                                                                                                                                                           ⊳ Set of edges  5:  for i ← 1 to N do  6:       select position p from P                                                                                                                                                                                               ⊳ Sequential selection  7:       α ← empty list of size A                                                                                                                                                                                                 ⊳ List of attributes  8:       for each ai in α do  9:           ai ← normal(μ = 0,σ2 = 1) 10:       end for 11:       create node v with position p and attributes α 12:       add v to V 13:  end for 14:  D ← empty dictionary                                                                                                                                                                                                              ⊳ Dictionary of distances between nodes 15:  for each v in V do 16:       for each u in V ∖{v} do 17:           d ←| p(v) − p(u) | /L                                                                                                                                                                          ⊳ Normalized distance between v and u 18:           add (u,d) to D[v]                                                                                                                                                                                                                                            ⊳ Store distance 19:       end for 20:       sort D[v] by distance in ascending order 21:  end for 22:  nodeGroups ← findDivisors(N)                                                                                                                                                                                                                ⊳ Power law node distribution 23:  edgeCounts ← reverse(nodeGroups)                                                                                                                                                                                                          ⊳ Power law edge distribution 24:  for i ← 1 to length(nodeGroups) do 25:       numNodes ← nodeGroups[i] 26:       numEdges ← edgeCounts[i] 27:       selectedNodes ← select numNodes nodes from V 28:       for each v in selectedNodes do 29:           potentialEdges ← first numEdges nodes from D[v] 30:           for each u in potentialEdges do 31:                s ←∑A     j=1 v(αj) × u(αj)                                                                                                                                                                                                                                ⊳ Dot product 32:                if s > 0 and D[v][u] ≤ t then 33:                     create edge e between v and u 34:                     if e / ∈ E then 35:                          add e to E 36:                     end if 37:                end if 38:           end for 39:           remove v from V                                                                                                                                                                                                        ⊳ Node processed 40:       end for 41:  end for 42:  return Graph G(V,E) 43:  function FINDDIVISORS(n) 44:       divisors ← empty list 45:       while n > 1 do 46:           add ⌊n/2⌋ to divisors 47:           n ← n −⌊n/2⌋ 48:       end while 49:       return divisors 50:  end function___________________________________________________________________________________________________________________________________________________________________________________________________________________________________________

Results and Discussion

Figure 2 contains four generated graphs using the ARCANE algorithm. Visually, it is easy to notice that few nodes are central to the graph. It can also be seen that there are nodes without any edges. Even though the plots hint at the power law distribution of edges, a more detailed and thorough analysis is provided in the following sections.

Figure 2 Generated graph examples.

Our original grid layout approach is used to draw the graph nodes and edges. Connected components are denoted with different colors.

Experimental environment

We used the programming language Python version 3.10 and PyCharm integrated development environment for the development of ARCANE and for conducting our experiments. We imported csv, pickle, networkx, cdlib, matplotlib, numpy, sklearn and other necessary libraries for compared network generators for the implementation of our algorithm. A computer with the following specifications is used for simulating the generation process for the evaluation of our algorithm: Intel-i7 8-core processor, 32GB of memory, Nvidia 1070 GTX graphics card processor with 8GB of memory.

Evaluation of graph properties

We have given a list of required graph properties in earlier Required Graph Properties section. We have evaluated if the graphs generated by ARCANE conforms to these requirements.

Homophily property

For evaluating the homophily property in the generated graphs, the metric explained in Easley & Kleinberg (2010) is employed. This metric is also used in ANC generator (Largeron et al., 2015) .

The metric uses probability to quantify the homophily property in a given graph with attributes. In its existing applications, the metric uses a single attribute to calculate the homophily measure (HM). Since ARCANE can use more than one attribute, the metric has been expanded to accommodate them by summing HM for each attribute, then normalize it so that it is between 0 and 1. The normalized homophily measure (NHM) value for the generated graphs are listed in Tables 3 and 4.

Table 3 Model properties and assumptions.

Average (Avg.) measurements come from 10 simulations performed with the same graph properties as A = 10.

Measures (Average)	N = 250	N = 500	N = 1k	N = 2k	N = 2.5k	
Clustering coeff.	0.184 (0.022)	0.189 (0.018)	0.178 (0.014)	0.174 (0.011)	0.138 (0.011)	
Degree	3.364 (0.260)	3.848 (0.262)	4.639 (0.294)	5.163 (0.233)	3.729 (0.170)	
Betweenness centrality	7.15e−3(2.83e−3)	2.25e−3(3.31e−4)	9.96e−4(9.61e−5)	5.61e−4(1.42e−4)	2.04e−4(2.316e−5)	
Closeness centrality	0.066 (0.008)	0.049 (0.004)	0.036 (0.007)	0.027 (0.005)	0.015 (0.006)	
Edge count	420 (33)	962 (66)	2,319 (147)	5,163 (233)	4,661 (213)	
Conn. Component count	57 (5)	130 (7)	277 (13)	574 (18)	917 (27)	
Diameter	18 (4)	19 (2)	25 (2)	28 (3)	31 (8)	
SPL	5.900 (1.115)	5.507 (0.268)	6.126 (0.425)	7.370 (0.891)	7.150 (1.070)	
NHM	0.891 (0.016)	0.876 (0.006)	0.862 (0.008)	0.858 (0.008)	0.855 (0.005)	

Table 4 Model properties and assumptions.

Avg. measurements come from 10 simulations performed with the same graph properties as N = 500.

Measures (Average)	A = 10	A = 25	A = 50	A = 75	A = 100	
Clustering coeff.	0.189 (0.018)	0.171 (0.016)	0.159 (0.021)	0.169 (0.014)	0.155 (0.012)	
Degree	3.848 (0.262)	3.771 (0.217)	3.794 (0.232)	3.833 (0.146)	3.836 (0.213)	
Betweenness centrality	2.25e−3(3.31e−4)	2.48e−3(6.92e−4)	2.34e−3(5.12e−4)	2.43e−3(3.98e−4)	2.43e−3(3.59e−4)	
Closeness centrality	0.049 (0.004)	0.048 (0.004)	0.051 (0.005)	0.052 (0.005)	0.050(0.005)	
Edge count	962 (66)	942 (54)	949 (58)	958 (37)	959 (53)	
Conn. Component count	130 (7)	137 (11)	137 (8)	137 (7)	134 (10)	
Diameter	19 (2)	19 (3)	19 (1)	19 (3)	18 (3)	
SPL	5.507 (0.268)	5.806 (0.700)	5.504 (0.358)	5.569 (0.329)	5.668 (0.415)	
NHM	0.876 (0.006)	0.847 (0.009)	0.832 (0.007)	0.820 (0.008)	0.819 (0.007)	

Power-law edge generation

The generated graphs use a power-law degree distribution during their edge generation process. The results are compared with a network generated by the LFR benchmark, which also uses power-law degree distribution (Lancichinetti, Fortunato & Radicchi, 2008). As can be seen from Fig. 3, both LFR and ARCANE display the characteristic properties of the power-law distribution; fewer number of nodes have greater number of edges, while most have fewer.

Figure 3 ARCANE vs. LFR for edge degree distributions. They both use power-law degree distributions.

General properties of the model

Average clustering coefficient (CC) is a measure of how tightly the nodes in a network cluster together (Watts & Strogatz, 1998). The average number of steps along the shortest paths for all potential pairs of nodes is known as average shortest-path length (SPL) (or characteristic path length) in networks. It is a way to assess how effectively the information transports over a network. Average degree gives the average node degree in the whole network. The clustering coefficient, the average shortest path length and the average node degree are three most reliable metrics of a network’s topology. The graph diameter is the average eccentricity of the nodes in a graph, which the node eccentricity may be described as the longest distance from given node to all other nodes.

We also used centrality measures for evaluation. The total percentage of all-pairs shortest paths that pass through a node represents its betweenness centrality (Brandes, 2001). The ratio of the average shortest path distance to a node over all reachable nodes is its closeness centrality (Freeman, 1978). The average of these two centrality values are used for evaluating the generated graphs.

When the categorization of is applied; CC, average degree, betweenness and closeness centrality measures give the characteristics of the node distributions (Xiang et al., 2021). Edge count, diameter and SPL measures are considered as general graph statistics.

The resulting measurements for the generated graphs are listed in two tables. Table 3 lists graphs with variable number of nodes (250 ≤ N ≤ 2,500) and constant number of node attributes (A = 10). Generated graphs with constant number of nodes (N = 500) and variable number of node attributes (10 ≤ A ≤ 100) are given in Table 4.

We also have a separate result for an example generated graph with N = 104, A = 20, as (Clustering Coeff. = 0.121, Degree = 4.564, Betweenness Centrality = 5.73e-5, Closeness Centrality = 0.011, Edge count = 22,819, Conn. Component Count = 3,716, Diameter = 40, SPL = 8.3, NHM = 0.824). Due to the runtime issues (e.g., this particular simulation is lasted 1 h 48 min 11 s to be exact), we do not have enough simulation data for that many nodes to calculate the mean values, thus we have listed this result separately.

Running time evaluation

Table 5 lists the maximum running times out of 10 simulations for every different generation parameter used. The results show that the running times increase as the number of nodes and the number of attributes increase, as expected.

Table 5 Maximum runtimes in seconds for given graph properties out of 10 simulations.

Properties	A = 10	A = 25	A = 50	A = 75	A = 100	
N = 250	0.22	0.39	1.27	2.04	4.05	
N = 500	0.96	1.45	3.20	5.55	9.54	
N = 1000	7.74	9.02	12.60	17.91	26.48	
N = 2000	51.16	54.05	65.90	81.98	102.27	
N = 2500	94.95	105.20	115.65	126.63	141.09	

Comparison with other graph generators

ARCANE is compared to other generators with respect to several evaluation metrics. First, it is compared to ANC (Largeron et al., 2015), DANCer (Benyahia et al., 2016), GenCAT (Maekawa et al., 2023) and X-Mark (Citraro & Rossetti, 2021) with respect to their model properties. These generators were selected because they create graphs with communities and node attributes, they use similar approaches for their generation process and the source codes, or the applications of their algorithms were accessible for comparison. The resulting graphs from all these generators are passed to various community detection methods.

Model properties similarity evaluation

To compare ARCANE to other generators with respect to their model properties, graphs are generated by setting N = 500 and A = 10 for all. The results are listed in Table 6.

Table 6 Model properties comparison between graph generators.

Average measurements come from 10 simulations performed with the same graph properties for each graph as (N = 500, A = 10). Bold values indicate the most similar results.

Measures (Average)	ANC	DANCer	GenCAT	X-Mark	ARCANE	
Clustering coefficient	0.108 (0.025)	0.158 (0.026)	0.547 (0.043)	0.109 (0.017)	0.189 (0.018)	
Degree	4.492 (0.384)	8.248 (0.396)	29.558 (1.290)	11.465 (0.414)	3.848 (0.262)	
Betweenness centrality	7.11e−3(2.88e−4)	4.83e−3(4.26e−4)	2.26e−3(2.20e−5)	3.61e−3(1.46e−4)	2.25e−3(3.31e−4)	
Closeness centrality	0.224 (0.007)	0.296 (0.021)	0.476 (0.002)	0.360 (0.010)	0.049 (0.004)	
Edge count	1,123 (96)	2,074 (96)	7,389 (322)	2,866 (104)	962 (66)	
Number of connected components	1	1	1	1	130 (6.867)	
Diameter	9.3 (0.458)	6.1 (0.539)	4	4.875 (0.331)	19.1 (2.385)	
Average SPL	4.539 (0.144)	3.420 (0.217)	2.125 (0.011)	2.796 (0.073)	5.507 (0.268)	

From the results, it can be concurred that, according to the model properties, ANC generator is the one that has the most resemblance to ARCANE. From eight measures, five of results for ANC is very similar to our method’s results. In one measure, specifically in clustering coefficient, the most similar generator is DANCer. Also, only in betweenness centrality measure, the result of GenCAT generator is almost identical.

Community detection evaluation

Unknown ground truth communities.

When ground-truth communities are not known, then different metrics can be used to evaluate the community quality of the graphs. The Silhouette coefficient is an example of such evaluation metric, where a higher Silhouette Coefficient score relates to a model with better defined communities (Rousseeuw, 1987). It is a score between −1 and +1, where the positive scores near to +1 means that the graph is densely clustered. Near-zero scores can indicate overlapping communities.

Another metric is Calinski-Harabasz score, also known as the variance ratio criterion, where a higher score relates to a model with better defined communities (Caliński & Harabasz, 1974). The Davies–Bouldin index can also be used to evaluate the graph models, where a lower index relates to a model with better separation between the communities (Davies & Bouldin, 1979). The lowest possible score for Davies–Bouldin index is zero, which indicates a better partitioning.

For all these metrics, the number of clusters must be predetermined to measure the community quality. We used more than one number for the cluster count to evaluate the effects of this change. The mentioned metrics are applied to the results of a K-means clustering algorithm. We used the node attribute values list per graph node for the training instances to cluster with K-means algorithm.

It can be seen from Table 7, that the Silhouette coefficient increases in direct proportion to the number of predefined number of labels. As it is very close to 0, it can be speculated that the communities in our generated graphs may be defined as overlapping communities. Calinski-Harabasz score and Davies–Bouldin index are decreasing with the increasing number of labels. Combining these three results, the connected components in our generated networks are assumed to be the ground-truth communities for the next evaluation.

Table 7 Evaluation results for ARCANE when ground truth communities are assumed to be not known.

CC is the connected component count of the given graph (N = 500, A = 10).

K-means label count =	3	A/2	A	CC	
Silhouette coefficient	0.075 (0.002)	0.075 (0.003)	0.081 (0.003)	0.090 (0.004)	
(−1 < sc < 1)	
Calinski & Harabasz score	40.422 (1.353)	34.312 (1.090)	26.933 (0.414)	8.464 (0.140)	
(0 ≪ ch_score)	
Davies & Bouldin index	2.859 (0.055)	2.472 (0.053)	2.117 (0.033)	1.222 (0.041)	
(0 ≤ db_index)	

Connected components as ground truth communities.

Following the results of previous metrics, assuming connected components as the ground-truth communities comes from the fact that in social relations, being in a community means that being highly connected to other nodes (Yang & Leskovec, 2012).

These ground-truth communities are compared to the communities detected by the CD algorithms. I-Louvain (Combe et al., 2015) and EVA (Citraro & Rossetti, 2019) were selected for attribute-aware CD methods to detect the attribute-aware communities. General CD methods which do not consider the node attributes while detecting the communities have also been used, such as CPM (Palla et al., 2005), Louvain (Blondel et al., 2008), and APAL (Doluca & Oğuz, 2021). These methods have been chosen to find out if the structural communities detected by general CD methods will have a similarity score more than the ones that are found by attribute-aware CD methods. According to the results of this evaluation, we also aim to select the attribute-aware CD method that can be used in community comparisons with other generators. The metrics, the Normalized Mutual Index (NMI) and the Adjusted Rand Index (ARI), are used for similarity evaluation.

For both NMI and ARI, the lower and upper bounds are 0 and 1, respectively. Values closer to zero mean that there is not any similarity between the given communities and the communities detected by the CD method, while values closer to one mean that they are similar. The results are listed in Table 8.

Table 8 Evaluation results for ARCANE ground truth communities compared to the communities found by CD methods (N = 500, A = 10).

Bold values indicate the most successful results.

Measures:	NMI	ARI	
CPM	0.785 (0.021)	0.354 (0.069)	
Louvain	0.848 (0.024)	0.205 (0.044)	
APAL	0.257 (0.041)	0.211 (0.079)	
I-Louvain	0.890 (0.009)	0.310 (0.038)	
EVA	0.708 (0.05)	0.128 (0.091)	

From the results, it can be seen that according to NMI metric, I-Louvain is the one that finds most similarities. The close second in this metric is Louvain method. In ARI metric, CPM is the one that finds two different community list, ground-truth and detected, as most similar. In this metric, the second score again comes from I-Louvain method.

Under the light of these results, the community similarity of our generator is evaluated with I-Louvain, the most successful attribute-aware CD method.

Community similarity comparison.

I-Louvain (Combe et al., 2015) is selected as the attribute-aware CD method to compare the communities formed by ARCANE and other generators. For the community similarity evaluation metric, NMI is chosen, since it is commonly employed. The results are listed in Table 9.

Table 9 Similarity evaluation for ARCANE in comparisons with other graph generators using node-attribute aware CD methods.

Bold values indicate the most successful results.

CD methods:	I-Louvain	
Measure:	NMI	
ANC	0.1 (0.076)	
DANCer	0.159 (0.118)	
X-Mark	0.103 (0.052)	
GenCAT	0.048 (0.018)	

It is evident in these results that the most similar generator to ARCANE is DANCer with respect to the detected community aspect. NMI results of I-Louvain CD method also show that X-Mark and ANC also have subjectively high results. The least similar generator in detected communities is the GenCAT generator. When the results of this evaluation and the evaluation of model properties similarity are combined, it is seen that GenCAT generator has less similarities to ARCANE than ANC, DANCer and X-Mark. Therefore, we decided to remove GenCAT in our follow-up evaluations.

Evaluation with a real dataset

For evaluating our generator further, a real network dataset called Sinanet is used (Jia et al., 2017). It is a microblog user relationship network with 3,490 nodes, 30,282 edges and 10 numerical attributes per-node.

There are two parts for this evaluation. In the first part, the model properties with respect to Sinanet network are evaluated. In the second part, it is evaluated if the communities generated in our model are similar to the ones in Sinanet network.

Model properties evaluation

Networks with the same number of nodes and node attributes with Sinanet are generated using ARCANE, where N = 3, 490 and A = 10 for every node-attributed graph generator. These generators, ANC, DANCer, X-Mark and ARCANE, are compared using maximum mean discrepancy (MMD) over clustering coefficient, degree, betweenness and closeness centrality values, and by using absolute mean difference (AMD) over edge count, connected component count, diameter and average shortest path length values for their model properties. The result for this evaluation is given on Table 10.

Table 10 Model properties comparison between graph generators using Sinanet benchmark.

Measurements come from simulations performed with the same graph properties for each graph as (N = 3490, A = 10). ANC and DANCer graphs are generated in their respective applications and exported into our environment for comparison. Bold values indicate the most successful results.

MMD over	ANC	DANCer	X-Mark	ARCANE	
Clustering coefficient	0.043	0.035	0.014	0.004	
Degree	0.078	0.013	0.041	0.119	
Betweenness centrality	8.53e−07	3.67e−07	5.91e−08	2.55e −08	
Closeness centrality	0.044	0.026	0.006	0.233	
AMD over	
Edge count	19,647	14,463	8,046	19,557	
# of Connected components	23	23	23	999	
Diameter	5	1	0	38	
Average SPL	2.513	1.236	0.563	7.359	

From the results of the model evaluation, it can be said that the most similar graphs to Sinanet are generated by our generator, ARCANE. X-Mark is also as successful as our method on three different measures, closeness centrality, edge count and diameter. The most similar graphs on degree measure are generated with DANCer generator.

When the connected component count measure is considered, it can be seen that all other generators have an AMD value as 23. This is due to the fact that all other generators create connected graphs. However, the graphs generated by ARCANE are disconnected graphs. The AMD over connected component count is higher in our method than in any other generator, as shown in the table, but it is still identified it as the most successful one because the Sinanet network is also a disconnected graph.

Community detection evaluation

To evaluate the communities formed by ARCANE, the attribute-aware CD method, I-Louvain, is used to detect the communities in Sinanet, and to compare these detected communities with ARCANE and the existing generators. NMI measure is used to evaluate the community similarity. Results of this comparison is given on Table 11.

Table 11 Community detection comparison between graph generators using Sinanet benchmark.

Measurements come from simulations performed with the same graph properties for each graph as (N = 3, 490, A = 10). ANC and DANCer graphs are generated in their respective applications and their data are exported into our environment for comparison. Bold values indicate the most successful results.

CD methods:	I-Louvain	
Measure:	NMI	
ANC	0.009 (0.002)	
DANCer	0.013 (0.002)	
X-Mark	0.009 (0.005)	
ARCANE	0.065 (0.003)	

From the results, it can be said that when the communities are detected by the I-Louvain method, the graphs generated by ARCANE has the most similar communities to the Sinanet real dataset. The other generators that were evaluated exhibited fewer similarities to Sinanet communities than our method. Additionally, the difference between ARCANE and the other generators was nearly five or six times as great according to the NMI metric.

It can be argued that our method creates networks that have real-world network properties on both model similarity and the community aspects. Therefore, ARCANE can be used safely in cases where synthetic social networks are needed.

Limitations

Even though the algorithm generates graphs as aligned with our intentions, there are some limitations that should be addressed and opportunities to improve in the future.

ARCANE currently only generates undirected unweighted graphs. It is a limitation in many aspects, such as the lack of directed information propagation in social networks can be a missing functionality in a network generator (Schweimer et al., 2022). If we think that the nodes as people in a social network, they may choose to share information with another person in one-way in certain cases. Some sensitive information is kept hidden from the public, or in this case, the community the person belongs to. We created the affinity levels of the node attributes with this mindset, but there can be some public and some private node attributes by taking this knowledge into consideration. This intimacy between individuals is an aspect yet to be covered in future for better graph generators.

Another limitation can be the node placement algorithm. We used diamond-square algorithm to place the nodes in our generated graphs. Yet, in this usage, we do not place the nodes considering the proximity between nodes, the node similarities, or the power-law edge generation aspect. Placing more central nodes to the correct places in the graph can create more connected graphs, or better community structures. Depending on this same situation, some nodes do not have any edges as shown in Fig. 2. Provided that these nodes are placed into the graph with better positions, these problems may be overcome.

One of the most important limitations of our study was the long running time of our algorithm. Since we create a graph generator, time complexity of it is also very important. Our first aim was to create an algorithm which generates graphs with communities and node attributes, and because of that, the running time was the least of our concerns at first. Using parallel-processing in the generation of synthetic graphs such as is an efficient solution in most cases, and ARCANE can make use of parallelism in the future (Bressan et al., 2013).

Our algorithm creates static graphs. However, in most cases, the graphs and the community structures in them change over time or over different factors. One of the articles that have been surveyed actually created their generator with dynamic communities in mind (Benyahia et al., 2016). Hence, the merge and split interactions between the communities must also be covered. Moreover, the node attributes are also subject to change if we think of individuals. People tend to dislike some things which they like in the past, or vice versa. Their specific traits can also change with time. An algorithm which also considers these changes can be very beneficial for creating a dynamic graph generator.

Generating large graphs can also come with other problems such as improper graph visualizations or unsuitable graph layouts. Also, there are not many research examples that show the contributions of the node attributes in their graph visualizations. These problems are also present in our work. We used our grid layout approach for showing the nodes at a better separated position, however, we accept that our approach has also its own shortcomings.

We can count the proximity and the node similarity calculations as our last limitations. For calculating the proximity, we benefited the physical proximity rules from social context (Stopczynski, Pentland & Lehmann, 2018). The humans can accept themselves near to or far from each other according to the place they are in. The effect of this assumption must be evaluated with respect to the usage area of our generator. Different use-cases can have their own proximity calculations. We solved this problem in our generator by using an external ratio value for the conceptual proximity; however, a fully different proximity calculation approach may be adopted, since it is a subjective solution. Measuring node similarity in accordance with the values of the node attributes may have its disadvantages, too. One can think that some of the node attributes influences the edge generation phase more than other attributes in the graph. In the future, implementing the node attributes with different importance levels or giving different weights to them may be a better approach.

Conclusions

We designed an algorithm to create synthetic networks with communities and node attributes. The algorithm can create the edges between the nodes with respect to the proximity and homophily aspects of social context. The number of parameters is few and the algorithm can be tweaked by defining functions to control the probability distributions for the attribute values. Our algorithm, ARCANE, is available online (https://github.com/koguz/arcane/).

The algorithm has been evaluated with different generation parameters to estimate its significance. The properties of the generated graphs are compared to earlier similar network generators such as LFR, ANC, DANCer, X-Mark and GenCAT. We evaluated how the node attributes and edge generation process affect the communities by using both general and attribute-aware CD methods such as CPM, Louvain, APAL, I-Louvain and EVA. We also evaluated the resulting graphs against a real-world network, Sinanet. Our evaluations suggest that we created a successful social network generation algorithm, compared to earlier similar and accomplished variants.

One of the most important ideas we consider while we are creating ARCANE is the broad spectrum of its usage areas. We employed it mostly in social context; however it can be applied to different networks with minimum adjustments. ARCANE is a robust, easy to use and reliable synthetic graph generator with communities and node attributes and it is implemented in one of the most used programming languages nowadays with object-oriented programming principles. We hope that ARCANE and further iterations of it can be used as a base network generation solution for various applications in AI, computer games, biology and communications.

The authors would like to thank the journal editors and reviewers for their invaluable feedback.

Additional Information and Declarations

Competing Interests

Author Contributions

Data Availability

The authors declare there are no competing interests.

Muhtar Çağkan Uludağlı conceived and designed the experiments, performed the experiments, analyzed the data, performed the computation work, prepared figures and/or tables, authored or reviewed drafts of the article, and approved the final draft.

Kaya Oğuz analyzed the data, prepared figures and/or tables, authored or reviewed drafts of the article, and approved the final draft.

The following information was supplied regarding data availability:

The source code for our method is available at GitHub and Zenodo:

- https://github.com/koguz/arcane.

- Uludağlı, M. Ç., & Oğuz, K. (2023). Attribute-based Realistic Community and Associate NEtwork (ARCANE) generation algorithm. Zenodo. https://doi.org/10.5281/zenodo.13933537.

The Sinanet dataset is available at GitHub:

- https://github.com/smileyan448/Sinanet.

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
