# Peer review of "From attributes to communities: a novel approach in social network generation"

_PeerJ Computer Science, doi:10.7717/peerj-cs.2483_

## Round 0.1 · original submission · Minor Revisions

The manuscript "From attributes to communities: A novel approach in social network generation" presents an original contribution and is generally within the scope of PeerJ Computer Science. Both reviewers have provided favorable feedback, recognizing the well-structured and clear presentation of the work, the detailed methodology, and the valid experimental design. However, both reviewers agree that additional work is needed before the manuscript can be accepted for publication.

Reviewer 1 suggests improvements to the abstract, specifically to clarify the abbreviation "ARCANE" and to provide a more detailed description of the proposed algorithm and the results. Furthermore, the addition of 4 to 6 keywords is recommended. Reviewer 1 also requests that the proposed algorithm be presented in pseudocode, along with a more detailed explanation. Additionally, the experimental environment should be described more comprehensively, including the operating system, programming language, and tools used. The conclusion section should be shortened, and some sentences should be moved to other sections of the paper. The authors are also encouraged to revise the manuscript for grammar, spelling, and formatting issues. A few additional references are suggested to strengthen the related work, although its inclusion is left to the authors' discretion.

Reviewer 2 commends the well-prepared related works section and the clear articulation of the algorithm and methodology. However, improvements to the abstract are recommended to better introduce the problem and the proposed solution. Minor punctuation issues are noted, and the authors are asked to update their GitHub repository with clearer usage instructions and examples, including sample data for easier implementation. Additionally, a reference is requested for line 121, although its inclusion again is left to the authors' discretion.

·

Basic reporting

Comments on the attached file.

Experimental design

Comments on the attached file.

Validity of the findings

Comments on the attached file.

Additional comments

Comments on the attached file.

Reviewer 2 ·

Basic reporting

- The first sentence of the Abstract should introduce the problem the proposed solution addresses, followed by an explanation of what is being proposed.
- The Related Works section is well-prepared with maintained coherence. Lines 116 and 117 could be combined into a single paragraph, as the split is unnecessary.

- Line 121 requires a reference for the sentence. Suggested citation: Moustakas, L. (2023). Social Cohesion: Definitions, Causes, and Consequences. Encyclopedia, 3(3), 1028-1037.

- Punctuation issues: On lines 157 and 161, the colon (':') is missing. Please add it for consistency.

Experimental design

- The algorithm and methodology are well-defined, with steps clearly articulated.
- The design and the reported experimental values are valid and detailed.

Validity of the findings

- The limitations are clearly defined, providing a foundation for potential future improvements.
- The conclusion effectively ties together the paper, supported by the presented results.
- The source code has been shared for transparency. However, I would ask for updating the GitHub repository with clearer usage instructions and examples, including sample data for easier implementation before accept.

Cite this review as

---

## Round 0.2 · accepted · Accept

In my opinion the work is now ready for publication